

# Modelling of climate change impact on flow conditions in the lowland anastomosing river

Paweł Marcinkowski and Dorota Mirosław-Świątek

Department of Hydrology, Meteorology and Water Management, Warsaw University of Life Sciences—SGGW, Warsaw, Poland

Corresponding author
Paweł Marcinkowski,
p.marcinkowski@levis.sggw.pl

## ABSTRACT

The progressive degradation of freshwater ecosystems worldwide requires action to be taken for their conservation. Nowadays, protection strategies need to step beyond the traditional approach of managing protected areas as they have to deal with the protection or recovery of natural flow regimes disrupted by the effects of future climate conditions. Climate change affects the hydrosphere at catchment scale altering hydrological processes which in turn impact hydrodynamics at the river reach scale. Therefore, conservation strategies should consider mathematical models, which allow for an improved understanding of ecosystem functions and their interactions across different spatial and temporal scales. This study focuses on an anastomosing river system in north-eastern Poland, where in recent decades a significant loss of the anabranches has been observed. The objective was to assess the impact of projected climate change on average flow conditions in the anastomosing section of the Narew River. The Soil and Water Assessment Tool (SWAT software) for the Narew catchment was coupled with the HEC-RAS one-dimensional unsteady flow model. The study looked into projected changes for two future time horizons 2021–2050 and 2071–2100 under the Representative Concentration Pathway 4.5 using an ensemble of nine EURO-CORDEX model scenarios. Results show that low flow conditions in the anastomosing section of the Narew National Park will remain relatively stable in 2021–2050 compared to current conditions and will slightly increase in 2071–2100. Duration of low flows, although projected to decrease on an annual basis, will increase for August–October, when the loss on anastomoses was found to be the most intense. Hydraulic modeling indicated extremely low flow velocities in the anastomosing arm (<0.1 m/s) nowadays and under future projections which is preferable for in-stream vegetation development and their gradual sedimentation and closure.

## INTRODUCTION

Climate change is considered to be one of the greatest challenges nowadays, and preventing it is a key strategic priority for the European Union. It is expected that it will

amplify current pressures on natural resources, but also create new ones (*IPCC, 2014*). Riverine ecosystems, which provide a diverse range of services, upon which humans are dependent, are listed amongst the most sensitive to climate change of all ecosystems (*Ormerod, 2009*). Thus, it is obvious that effective water management is crucial to successful climate change adaptation (*Ostfeld et al., 2012*). Climate change affects riverine ecosystems directly by determining hydrological processes (where precipitation, temperature and evaporation are the key drivers), and indirectly, by changing the human use of river catchments, riparian zones and floodplains (*IPCC, 2014*; *Ormerod, 2009*). In the face of the progressive degradation of riverine ecosystems worldwide (*Millennium Ecosystem Assessment, 2005*), communities and governments are forced to consider strategies for conservation management. Such strategies need to step beyond the traditional approach of managing protected areas, as they have to deal with protection or recovery of natural flow regimes, disrupted by effects of future climate change, which is considerably uncertain (*Kingsford, 2011*).

Conservation strategies and particular measures preventing further decline in the health of aquatic ecosystems should consider mathematical models, which allow for an understanding of ecosystem functions and their interactions across different spatial and temporal scales (*Jähnig et al., 2012*). Catchment-scale hydrological changes driven by climate change strongly impact conditions at finer scales that is, in-stream hydraulics in river reaches (*Kiesel et al., 2013*). Attempts to model the impact of climate change on water resources at watershed scale have been conducted extensively (*Giri, Arbab & Lathrop, 2019*; *Napoli, Massetti & Orlandini, 2017*; *Krysanova, Kundzewicz & Piniewski, 2016*; *Kundzewicz et al., 2018*; *Kharel & Kirilenko, 2018*; *Tamm et al., 2018*), however, such large scale models still lack components that allow downscaling catchment-scale hydrological processes outputs to in-stream hydraulics. Therefore, an Integrated Modelling Framework (IMF) has been proposed by *Jähnig et al. (2012)* aimed at providing the environmental data and describing highly complex interactions between atmosphere, hydrosphere and biosphere across different spatial scales. So far, such IMF approach has been successfully applied in numerous studies: *Guse et al. (2015)* simulated land use and climate change impact on hydraulics and habitat suitability for fish and macroinvertebrates; *Kail et al. (2015)* assessed the effect of different pressures on abiotic habitat conditions and the biota of rivers; *Visser, Beevers & Patidar (2019)* proposed a coupled hydrological and hydroecological modeling framework to assess the impact of climate change on hydroecological response. The scope of the presented studies (*Guse et al., 2015*; *Kail et al., 2015*; *Visser, Beevers & Patidar, 2019*) demonstrates the open-ended character of IMF which allows tackling diverse region and site-specific problems concerning riverine ecosystems.

One of the examples of such issues described by *Lewin (2010)* and *Walter & Merritts (2008)* is the loss of multichannel anastomosing rivers from most of the temperate lowland floodplains due to direct and indirect human activities. Once very common, now they are rare in the developed world, which requires their conservation even more. Most of the studies concerning the origin and evolution of anastomosing rivers were qualitative in nature (*Kleinhans et al., 2012*; *Makaske et al., 2017*), and thus, more quantitative and

model-based assessment is required. Previous studies on modeling the multi-channel river systems mostly concentrated on vegetation-flow interactions (*Schuurman, Marra & Kleinhans, 2013*; *Marcinkowski, Kiczko & Okruszko, 2018*) and sediment transport (*Nicholas, 2013*). However, none of the studies investigated the impact of climate change on flow conditions, by means of hydrological modeling, which is crucial in the light of long-term conservation programs for riverine ecosystems protection.

Anastomosing rivers extinction has been reported widely and is a common issue around the world (*Lewin, 2010*; *Walter & Merritts, 2008*). Their conservation is challenging due to multiple anthropogenic and natural stressors. In this study we use the example of the anastomosing river system in north-eastern Poland, to describe the potential impact of climate change on river flow characteristics and recommendations for conservation plans. In the anastomosing Narew significant loss of the anabranches has been observed (*Marcinkowski, Giełczewski & Okruszko, 2018*) in recent decades. The mechanisms controlling the loss were reported by *Marcinkowski, Grabowski & Okruszko (2017)* and attributed mainly to changes in flow conditions, determined by different factors at catchment and reach scale. In the strictest sense, they observed that low water levels in anabranches facilitated sediment deposition and the colonization of common reed, while reduced high flows minimized sediment mobilization. Encroaching vegetation reduced flow efficiency and channel capacity, which in turn created perfect conditions for river bed aggradation and further reed colonization. After a few years, the former anabranch was eventually overgrown by reeds and disappeared completely. Therefore, as indicated by *Marcinkowski, Grabowski & Okruszko (2017)*, low flow conditions are of special concern due to the fact that the loss of anabranches is recognized to be most intense during the summer season. In another study *Marcinkowski, Kiczko & Okruszko (2019)* conducted an ex-ante model-based assessment of conservation measures efficiency, proposed in the protection plan (*Mioduszewski, Napiórkowski & Okruszko, 2014*) of the anastomosing section of the river in the Narew National Park (NNP). In the protection plan the following conservation measures were proposed: (1) mowing of in-stream vegetation, (2) dredging of anabranches and (3) placement of submerged wooden weirs at the main channel redirecting part of the flow into anabranch. *Marcinkowski, Kiczko & Okruszko (2019)* indicated that the only way to maintain the anastomosing character of the river requires highly invasive engineering solutions (channel dredging and building water dams). However, before implementing any engineering solutions, which might cause serious ecological consequences in the protected area, a look ahead into potential flow changes caused by climate change projections is required. It might turn out that the predicted increased precipitation in this part of Europe (*Jacob et al., 2014*) could potentially reverse the anastomosing system degradation by increasing low flows in anabranches—the main factor responsible for channel extinction. Or contrarily, it might turn out that higher evapotranspiration rates due to temperature increase might counteract and even overcompensate the additional rainfall and create a negative climatic water balance during the summer. This highlights the importance of analyzing the seasonal variability of climate change, and not only mean annual changes. Both anthropogenic and natural factors control flow regime at the catchment scale. The impact of
anthropogenic stressors on anastomosing system planform has been extensively investigated in a study conducted by *Marcinkowski & Grygoruk (2017)*, and therefore, we focus on the climate change impact in this study.

Against this background, the objective of this article is to assess the impact of projected climate change on average flow conditions in the anastomosing section of the Narew River, with the focus on low discharges in the main channel and the anabranches, which are threatened with extinction. The Soil and Water Assessment Tool model (SWAT, *Arnold et al., 1998*) for the Narew catchment is used and coupled with the HEC-RAS one-dimensional unsteady flow model, built for the anastomosing section of the river (where channel extinction process is recognized to be in progress), including the main channel and the anabranch. The study looks into projected changes for two future time horizons within the 21st century (2021–2050 and 2071–2100) under the Representative Concentration Pathway (RCP) 4.5 using an ensemble of nine EURO-CORDEX model scenarios (*Jacob et al., 2014*).

# MATERIALS AND METHODS

## Site description

The Upper Narew is the sub-catchment of the largest Polish river basin—the Vistula (Fig. 1). It drains an area of 4,231 km$^2$ (of which 27% belong to Belarus) and is characterized by a flat relief with an average elevation of 152 m a.s.l. The prevailing type of soils in the catchment are sands and loamy sands, whereas heavy, impervious soils are rare. The land cover in the Upper Narew catchment is predominantly forested (43.6%) and agricultural (41%), whereas wetlands and grasslands occupy 16% of the area. The catchment is located in a temperate climatic zone with moderately warm summers (mean temperature in July 18 °C), cool winters (mean temperature in January −2 °C), and an annual average precipitation total of ca. 600 mm.

The Upper River Narew stretch, which was investigated in this study (Fig. 1C), is characterized by low-gradient (0.0002 m/m) and anastomosing pattern. It is situated in the Narew National Park (NNP) for which Natura 2000 sites (under both Bird and Habitat directives) were established. Within the NNP (6,810 ha), the river is characterized by a complex network of small interconnected channels (*Gradziński et al., 2003*). Vegetation cover within the NNP is dominated by early growth sedge and reed communities. In-stream vegetation densely overgrows channels during the summer season, but also survives the dormancy season to some extent, influencing the hydraulics of channels for the whole year. The flow regime (based on discharge data series from the Suraż gauging station spanning from 1951 to 2018) is characteristic for lowland snow/rainfall-fed rivers with clearly dominating spring flooding, sourced by snowmelt occurring most commonly in April. Summer floods caused by heavy rain events are significantly lover. The average yearly discharge for the analyzed period is 15.4 m$^3$/s, whereas extreme events (i.e., minimum and maximum flows) equal 1.5 m$^3$/s and 250 m$^3$/s, respectively. The number of days with flooding is diverse (from 0 to 119; 37 on average) and depends majorly on the snow cover thickness in winter season.

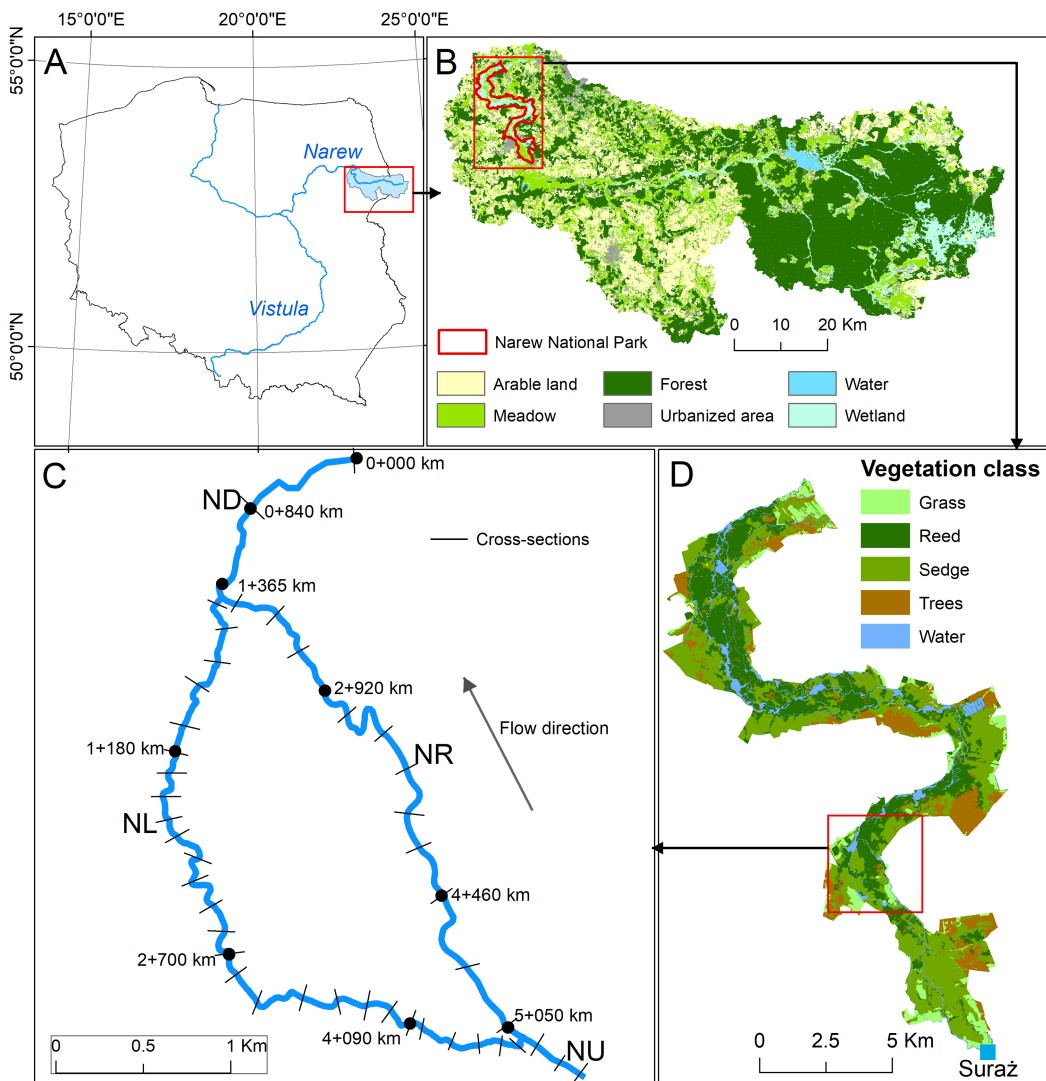

**Figure 1 Case study ((A) location, (B) land use, (C) river network, (D) vegetation type).** NU, Narew upstream reach; ND, Narew downstream reach; NL, Narew left reach; NR, Narew right reach.

This particular stretch has been selected for the analysis, as according to *Marcinkowski, Grabowski & Okruszko (2017)*, the channel extinction process was recognized to be in progress there for the last few years. It is worth noting that in current conditions during the growing season on average, nearly 98% of the flow is concentrated in the main channel (right-hand side reach on Fig. 1C) and only 2% is distributed to left-hand side anabranch. Such extremely low flows and unevenly distributed discharge create favorable conditions for vegetation encroachment and channel sedimentation, which in turn lead to the gradual closure of the side channels, as has been observed in the field for the last few years. Therefore, it has to be highlighted that if current hydrological conditions, especially low flows during the vegetation period, remain stable in the coming decades, the anastomosing system will most likely shift into a single-channel planform. The extinction

process is in progress now and it might be assumed that it will be continued unless the low flows increase.

## Integrated modeling framework

The IMF applied in this study consists of a catchment scale SWAT model and a river reach scale HEC-RAS model. The Hydrological SWAT model driven by climate change projections was used for daily discharge calculations for the future time horizons: 2021–2050 (near future—NF), and 2071–2100 (far future—FF). For a baseline scenario, which determines a reference point for the projected climate changes, historical (1971–2000) bias-corrected climate model data was used. In the study, flow condition changes were assessed at two different spatial scales: (1) at catchment scale mean monthly flows were calculated (for all SWAT sub-basins) showing the general impact of climate change on water resources in the area and (2) at reach scale daily hydrographs (implemented in 1D HEC-RAS hydraulic model as an upstream boundary condition and groundwater inflow from subcatchment) were calculated from the SWAT project. This way SWAT-based simulations of daily flow could be directly transferred and routed in the HEC-RAS stream network (Fig. 1C) as the flow hydrograph input, which allowed assessing the impact of climate change on hydraulic conditions in the anastomosing section of the River Narew. The projected flow hydrographs of models' ensemble as a SWAT model output can be found in the raw Supplemental Material (flow simulation files). As low flows prevail in summer and autumn months, triggering anabranch loss (*Marcinkowski, Grabowski & Okruszko, 2017*), HEC-RAS is only applied during these periods.

The resulting daily discharge was next reach-averaged and analyzed for flow velocity. Additionally, to assess not only the magnitude but also the duration of low flow conditions, hydrographs derived from SWAT were statistically examined. Given that the channel loss process is triggered by low flow conditions, a threshold value has been defined based on the daily discharge observations from Suraż gauging stations (1952–2017), expressed as $Q_{80}$ (with 80% exceedance probability). This particular exceedance probability has been selected for two reasons: (1) according to *Smakhtin (2001)*, flows within the range of 70–99% time exceedance are usually most widely used as design low flows, (2) collected discharge measurements used for HEC-RAS model calibration correspond to $Q_{80}$. Taking into account the $Q_{80}$ flow (7 m$^3$/s), for each climate model in the baseline, NF and FF horizons, the number of days lower than $Q_{80}$ were accumulated. Additionally, the mean value of flow of consecutive days below threshold was calculated. The analysis scheme is presented in Fig. 2.

## SWAT

SWAT is a process-based, continuous-time model which simulates the movement of water, sediment and nutrients on a catchment scale (*Arnold et al., 1998*). It is a comprehensive tool suitable for investigating the interaction between climate, land use and water quantity. It enables simulation of long-term impacts of climate changes on water, sediment and nutrient loads in catchments with varied topography, land use, soils and management conditions. Spatially, the catchment area is divided into sub-basins and

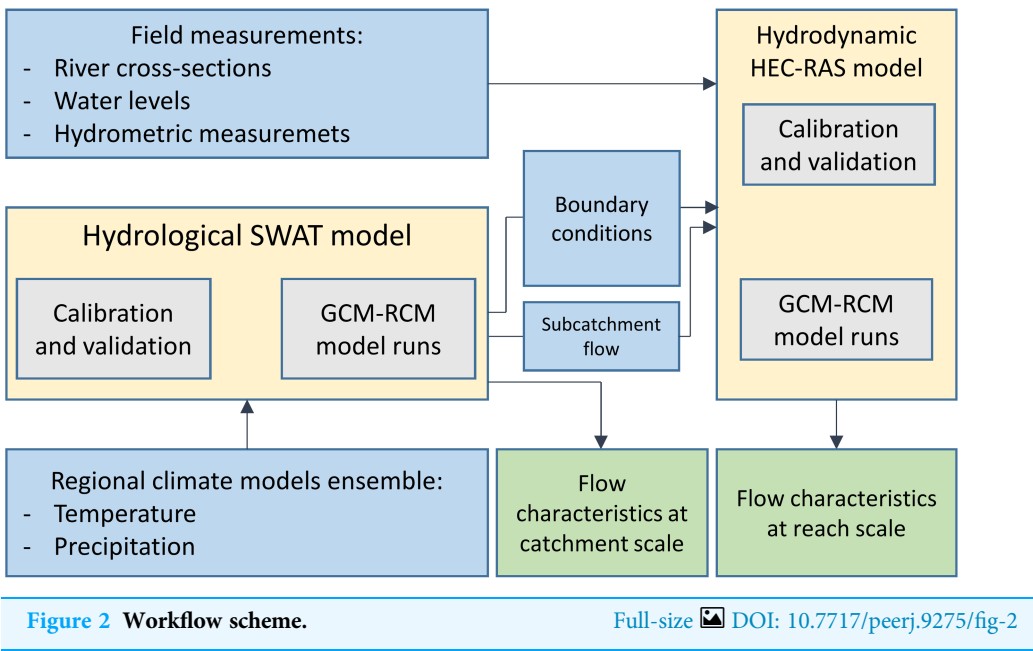

**Figure 2  Workflow scheme.**     

further into Hydrologic Response Units (HRU), which are the smallest spatially explicit units. Water balance as well as sediment and nutrient loads for each HRU are calculated at the land phase. Runoff is further aggregated to the sub-basin level and routed through the stream network to the main outlet in order to obtain the total runoff for the river basin. SWAT has also a simplified groundwater flow component which partitions groundwater into two aquifer systems: a shallow, unconfined aquifer which contributes return flow to streams and a deep, confined aquifer which contributes return flow to streams outside the watershed. In this study, we build upon the existing SWAT model of the Upper Narew catchment (*Marcinkowski et al., 2016*).

## Model setup, calibration and validation

Delineation of the catchment based on the 10-m resolution DEM resulted in division of the catchment into 243 sub-basins. The intersection of land cover map, soil map and slope classes resulted in creation of 4,509 HRUs. Daily precipitation and air temperature (minimum and maximum) data (1951–2013) were acquired from 5 km resolution gridded, interpolated using kriging techniques, dataset (CPLFD-GDPT5) based on meteorological observations coming from the Institute of Meteorology and Water Management (IMGW-PIB; Polish stations) (*Berezowski et al., 2016*). The use of interpolated climate data in the SWAT model, as reported by *Szcześniak & Piniewski (2015)* increased the model performance for a case study in Poland.

The calibration phase was conducted in SWAT-CUP using the SUFI-2 algorithm (Sequential Uncertainty Fitting Procedure Version 2) where the Kling–Gupta efficiency (KGE) was used as an objective function (*Gupta et al., 2009*). In the calibration and validation of the daily discharge, 10 flow gauges (data acquired from IMGW-PIB) were used. The calibration period was set to 1976–1985, and the validation period was 1986–1991, which was dictated by the highest number of observation records available in

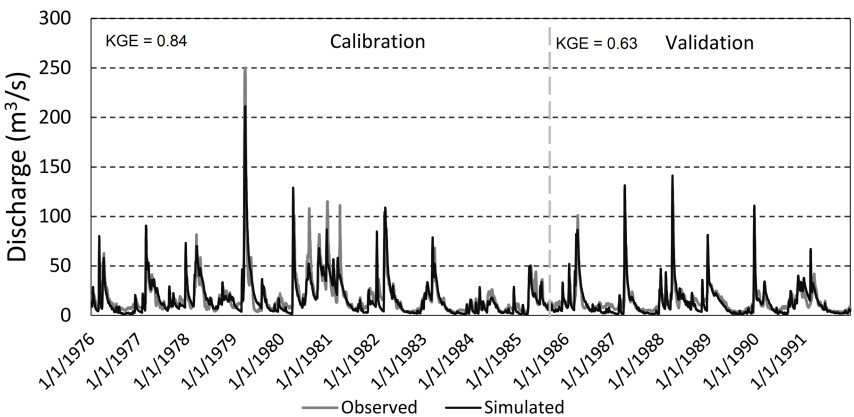

**Figure 3 Observed vs. simulated discharge in the Suraż gauging station in the calibration and validation period.**

the catchment (*Marcinkowski et al., 2016*). *Marcinkowski et al. (2016)* reported satisfactory values of goodness-of-fit measures across for all gauging stations (median KGE above 0.7). Figure 3 depicts the observed vs simulated daily discharge in the Suraż gauging station (cf. Fig. 1), which is at immediate upstream to the HEC-RAS model created for the anastomosing section of the Narew River.

### Climate change scenarios

In this study, SWAT was driven by climate forcing data from the CHASE-PL Climate Projections: 5-km Gridded Daily Precipitation & Temperature Dataset (*Mezghani, Dobler & Haugen, 2016*), consisting of nine bias-corrected General Circulation Model-Regional Climate Model (GCM-RCM) runs (involving four different GCMs and four different RCMs) provided within the EURO-CORDEX experiment projected to the year 2100 under RCP4.5 (*Piniewski et al., 2017*). All bias-corrected values (quantile mapping method by Norwegian Meteorological Institute (*Gudmundsson et al., 2012*)) of precipitation and air temperature were available for three time slices: 1971–2000 (historical period), 2021–2050 (near future—NF), and 2071–2100 (far future—FF). To deal with uncertainty caused by the use of different RCMs, a multi-model ensemble median approach was used.

### HEC-RAS

HEC-RAS is a well-known hydrodynamic model for one and two-dimensional hydraulic calculations and is often used in calculating both steady and unsteady, gradually varied flow (*Dysarz et al., 2019*; *Horritt & Bates, 2002*). In addition, the HEC-RAS system solves movable boundary sediment transport computations and water quality analysis. In 1D hydrodynamic component model solves the full 1D St. Venant equations for unsteady open channel flow with the finite difference method, using a four point implicit box scheme (*Brunner, 2016*). Manning's roughness coefficients are used to describe the flow resistance and were calibrated for the analyzed river branches. HEC-RAS performs hydraulic calculations for a full network of natural and constructed channels. Depending

on the type of reach junction (flow split or flow rejoin) in a river network, one equation of continuity of flow or equation continuity of stage is used as an interior boundary equation.

### Fieldwork and data collection

HEC-RAS (*Brunner, 2016*) model setup is strongly dependent on geometric and hydrological data. The geometric data consisted of river cross-sectional geometry collected at periodic, uniformly distributed (250 m intervals) stations along the selected, anabranching reach. The spacing of the measured cross-sections is related to Narew river topography (*Marcinkowski, Kiczko & Okruszko, 2017*). In the numerical model these cross-sections were used as a basic set for interpolating cross-section data at finer spacing. The spacing of measured cross-sections in the investigated river reach fulfills recommendation of *Samuels (1990)*, that distance between two cross-sections in 1D hydraulic model should be 10–20 times larger than bankfull surface width of a channel. In the analyzed river network, the average width of the main channels and the anabranches during the low flow condition is 17.61 m. Samuels recommendations were based on a combination of common sense, practical experience and mathematical equations. Additionally, results obtained by *Castellarin et al. (2009)* using 1D model HEC-RAS were consistent with the guidelines suggested by *Samuels (1990)*.

Cross-sectional data was collected using Real-Time Kinematic GPS (GPS-RTK) (Cross-sectional data available in raw Supplemental Material). Actual water surface profiles for investigated study reach at low flow rates was surveyed in the summer season. This was accomplished by surveying the water surface elevation at each cross-section using the GPS-RTK. Discharge was measured using a hand-held electromagnetic water flow meter at two river reaches (main channel and anabranch). The collected cross-sectional data, water surface level and discharge were used for model setup, calibration and validation.

### Model setup calibration and validation

The hydrodynamic modeling of the Narew River water flow was performed in the HEC-RAS model. In the hydrodynamic model, the river network was reproduced using four river reaches (cf. Fig. 1C). The Narew Upstream (NU) segment covers the upper section of the river from the cross section located at km 05 + 470 km to the split of flow at a channel junction to Narew right (NR) reach and Narew left (NL) reach, which represent the main Narew channel and its anabranch, respectively. Both sections are then combined in the Narew downstream (ND) segment (cf. Fig. 1C). In the model, the NU reach has 400 m of length and was represented through nine calculation cross-sections. The NR segment with a length of 3.69 km is represented by 75 cross sections, and the NL oxbow lake with a length of 4.715 km is described in the model with 95 cross-sections. The ND segment ending the analyzed river network is 1.365 m long and consists of 28 cross-sections. In the model, the average distance between calculation cross-sections is about 50 m. The upper boundary condition is formulated in the form of a flow hydrograph. The rating curve describes downstream boundary condition. The inflow from subcatchment calculated by SWAT, was applied in 1D hydraulic model as a hydrograph of uniform

lateral inflow. It can be assumed that in low flow condition, when a river has a draining function, this inflow describes river–groundwater interaction. The calibration and validation of the model was performed for the vegetation period, which differs significantly from the dormancy season in the abundance of in-stream vegetation. During the summer–autumn season a vegetation overgrows side anabranches causing high flow resistance and decreasing its capacity. The Manning's $n$ roughness coefficients were calibrated for four river reaches in a way that a unique roughness value has been assigned to the whole reach. In the calibration process a trial and error method has been used. The trial and error calibration involves repeated simulations while adjusting the model parameters until simulated results match measured variables. The calibration included the adjustment of Manning's coefficients for four branches so that the model results agreed with field measurement of water level and mean velocity. The initial Manning's coefficient were consistent with values estimated by *Marcinkowski & Grygoruk (2017)*. Following the initial set ($n_{NU} = 0.03$; $n_{NR} = 0.027$; $n_{NL} = 0.089$; $n_{ND} = 0.03$), the Manning's coefficients were manually adjusted to minimize the difference between the simulated and observed water level and the measured and simulated mean water velocity at nine measurement points in ND, NR and NL sections (cf. Fig. 1).

## RESULTS

### The catchment scale hydrological response to climate change

The projected mean annual temperature in the Upper Narew catchment is expected to increase by approximately 1.2 °C (ensemble range 0.8–1.6 °C) in NF and 2.1 °C (ensemble range 1.5–2.6 °C) in FF following the RCP4.5 according to the ensemble mean. Comparing the monthly variation, the highest change is expected to occur in November–March (2.4 °C, ensemble range 2.1–2.6 °C) and lowest in April–October (1.7 °C, ensemble range 1.5–1.9 °C) (*Marcinkowski et al., 2017*).

The annual total precipitation is projected to increase by 5.6% in the NF and by 9.5% in the FF, on the annual basis. The seasonal patterns show a relatively high increase in winter and spring, and a mixed response (both increase and decrease cases) in the summer and autumn. In the FF, the spring precipitation increase is distinctly higher than in other seasons, exceeding 20% (*Marcinkowski et al., 2017*) (Fig. 4).

The SWAT model simulations indicate that the median of projected changes in average water discharge, on an annual basis and catchment scale, is expected to increase by 11% and 25% in the NF and FF respectively. Monthly patterns showed the most pronounced increase occurring in December–March, with the ensemble median reaching 17% and 35% for NF and FF respectively. In the April–November period, a significantly lower increase in discharge is observed, reaching 7% and 23% for the NF and FF, respectively (Fig. 5). Additionally, more extreme and outlying values (both low and high) are noted for the future time horizons, compared to historical period. High flows might be attributed to the occurrence of extremely heavy rain events in the projected rainfall patterns, and low flows to an increased air temperature triggering the increase of evapotranspiration.

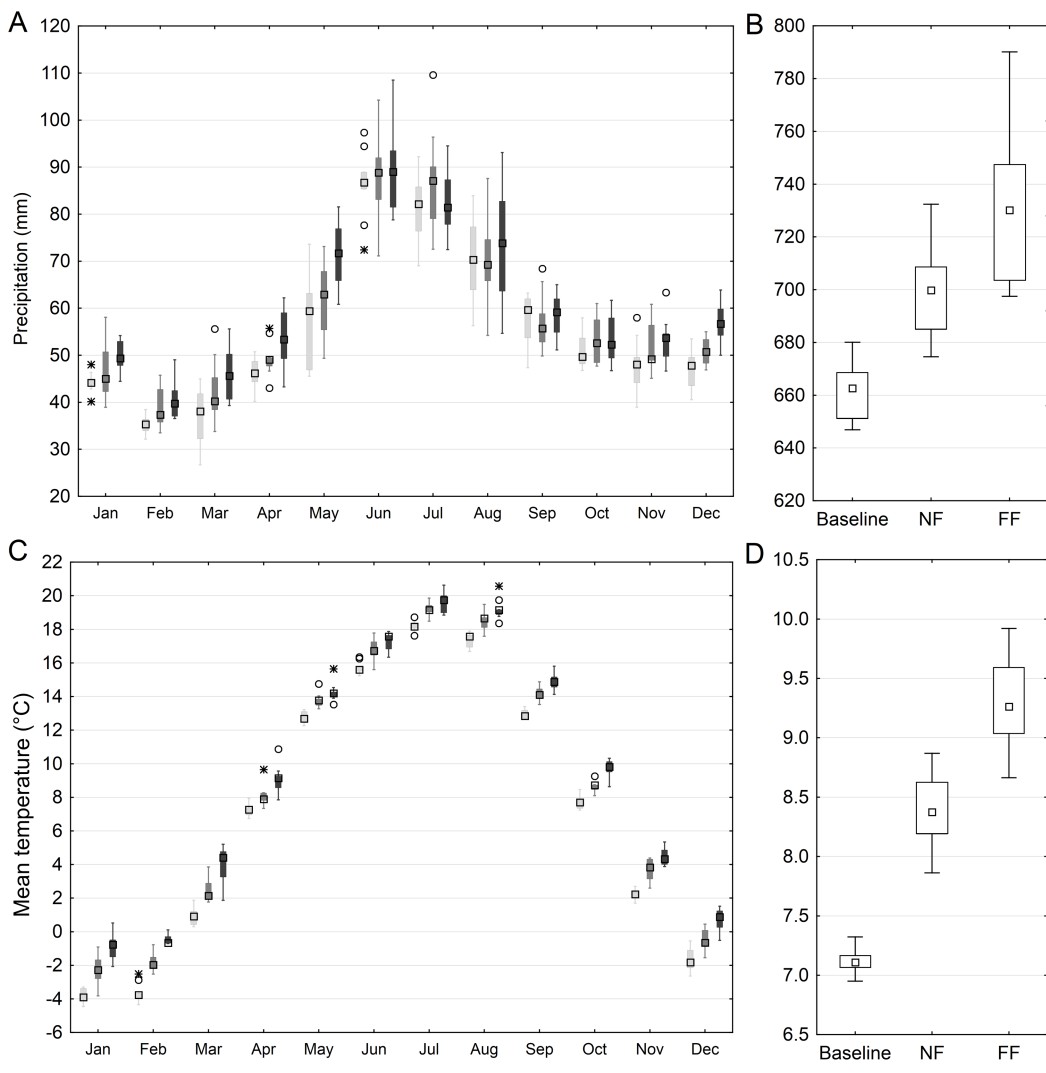

**Figure 4 Projected mean monthly (A) and annual (B) sum of precipitation and mean monthly (C) and annual (D) air temperature in the Upper Narew catchment.** Light gray color denotes baseline, medium gray—NF and dark gray—FF horizon; squares denote median values, boxes—1st to 3rd quartile values, whiskers—non outlier range, circles—outliers, asterisk—extremes.

## HEC-RAS model performance

A simulated split flow at the channel junction was compared with observations in the NR and NL reaches. In the calibration process for a discharge of 7.4 m$^3$/s, the error of the flow distribution is 0.7% and 11% for section NR and NL respectively. The average error of the water table in the river reaches varies from 0.03 m to 0.05 m. The mean error of simulated water velocity is 6%, 7% and 14% for the ND, NR and NL segments respectively (Table 1). The model accuracy discussed above was obtained for the following set of Manning's coefficients: $n_{NU} = 0.028$; $n_{NR} = 0.028$; $n_{NL} = 0.1$; $n_{ND} = 0.035$ m$^{-1/3}$/s. These values are in the range of the roughness coefficients used in other models for the analyzed section of the Narew River (*Marcinkowski, Kiczko & Okruszko, 2018*).

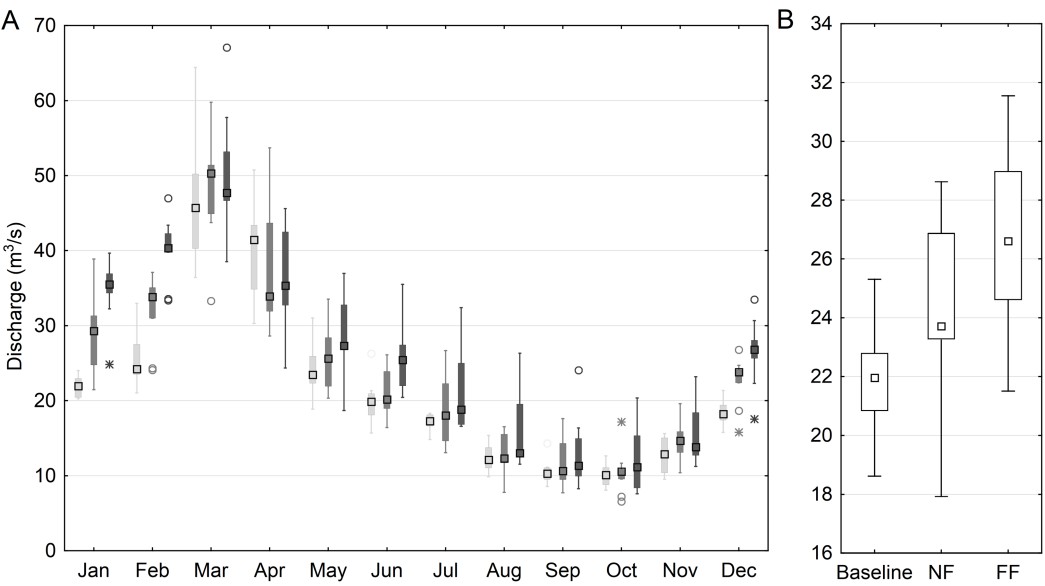

**Figure 5 Simulated mean monthly (A) and annual (B) discharge at the main outlet of the Upper Narew catchment.** Light gray color denotes baseline, medium gray—NF and dark gray—FF horizon; squares denote median values, boxes—1st to 3rd quartile values, whiskers—non outlier range, circles—outliers, asterisk—extremes.

**Table 1 Model errors (calibration run Q = 7.4 m³/s).**

| Reach | km | $Q$ (m³/s) | $Q_m$ (m³/s) | $E_Q$ (m³/s) | $H$ (m a.s.l) | $H_m$ (m a.s.l) | $E_H$ (m) | $V$ (m/s) | $V_m$ (m/s) | $E_v$ (m/s) |
|-------|----|----|----|----|----|----|----|----|----|----|
| ND | 0 + 000 | 7.4 | 7.40 | 0.0 | 115.18 | 115.16 | 0.05 | 0.43 | 0.46 | 0.02 |
|    | 0 + 840 |     |      |     | 115.38 | 115.42 |      | 0.41 | 0.43 |      |
|    | 1 + 365 |     |      |     | 115.48 | 115.57 |      | 0.40 | 0.42 |      |
| NR | 2 + 920 | 6.95 | 6.85 | 0.1 | 115.79 | 115.83 | 0.03 | 0.36 | 0.38 | 0.03 |
|    | 4 + 460 |     |      |     | 116.09 | 116.07 |      | 0.37 | 0.41 |      |
|    | 5 + 050 |     |      |     | 116.20 | 116.16 |      | 0.35 | 0.33 |      |
| NL | 1 + 180 | 0.45 | 0.55 | 0.1 | 115.66 | 115.71 | 0.04 | 0.04 | 0.05 | 0.01 |
|    | 2 + 700 |     |      |     | 115.89 | 115.93 |      | 0.07 | 0.08 |      |
|    | 4 + 090 |     |      |     | 116.10 | 116.06 |      | 0.05 | 0.06 |      |

**Note:**
Q, discharge; E, mean error in river reach; index m, model result; v, velocity; H, water level.

Verification of the developed model was based on measurements of the water level and water velocity performed at a discharge of 10.5 m³/s. In this case, the errors for the flow distribution are 0.9% and 9% for the sections NR and NL respectively. The average error of the water table varies from 0.04 m to 0.07 m. The mean error of simulated water velocity is 7%, 3% and 9% for the ND, NR and NL river reaches respectively (Table 2). The obtained errors are considered acceptable to allow the use of the model to simulate the unsteady flow during the vegetation period. Then, the developed hydrodynamic model was used to simulate the flow for the long historical period (1973–2000) and climate change scenarios for the years 2070–2100. In these simulations, the upper boundary

**Table 2 Model errors (validation run Q = 10.5 m³/s).**

| Reach | km | $Q$ (m³/s) | $Q_m$ (m³/s) | $E_Q$ (m³/s) | $H$ (m a.s.l) | $H_m$ (m a.s.l) | $E_H$ (m) | $V$ (m/s) | $V_m$ (m/s) | $E_v$ (m/s) |
|---|---|---|---|---|---|---|---|---|---|---|
| ND | 0 + 000 | 10.5 | 10.5 | 0 | 115.45 | 115.42 | 0.04 | 0.47 | 0.49 | 0.03 |
| | 0 + 840 | | | | 115.64 | 115.68 | | 0.44 | 0.47 | |
| | 1 + 365 | | | | 115.78 | 115.82 | | 0.41 | 0.45 | |
| NR | 2 + 920 | 9.55 | 9.64 | 0.08 | 116.10 | 116.07 | 0.07 | 0.40 | 0.42 | 0.01 |
| | 4 + 460 | | | | 116.36 | 116.30 | | 0.41 | 0.41 | |
| | 5 + 050 | | | | 116.49 | 116.38 | | 0.39 | 0.38 | |
| NL | 1 + 180 | 0.95 | 0.87 | 0.09 | 115.92 | 115.95 | 0.04 | 0.07 | 0.06 | 0.01 |
| | 2 + 700 | | | | 116.12 | 116.14 | | 0.10 | 0.09 | |
| | 4 + 090 | | | | 116.34 | 116.28 | | 0.09 | 0.08 | |

**Note:**
Q, discharge; E, mean error in river reach; index m, model result; v, velocity; H, water level.

condition in the form of a flow hydrograph was calculated in the SWAT hydrological model.

## Reach scale changes in flow condition

At the reach scale the flow condition changes were calculated based on the SWAT model simulations for the particular sub-basin, which spatially overlaps with the HEC-RAS hydraulic model. As indicated in Fig. 6, the duration of low flows changes significantly when comparing the historical period and future horizons. Considering the annual values (Fig. 6B), a gradual decrease from 151 (baseline) to 135 (NF) and 123 days (FF) is observed. However, the change is not uniform across the year and varies notably between each month (Fig. 6A). In general, for November–July, a clear drop in the duration of low flows in observed reaching 30% and 50% (ensemble median) for NF and FF respectively. However, for August–October, a slight increase in the duration of low flows is noted (8%) for the NF. Such a trend subsides in FF where again the decrease is observed, but significantly lower than for winter, reaching only 20%. The analysis of mean flow values of consecutive days below threshold indicate a gradual increase at the annual level from the historical period to future horizons (Fig. 6D). However, the change is not uniform across the year. In general, a higher increase is observed for FF compared to NF, and the highest increase occurs for February–May in both time horizons. What is noteworthy, the mean value of flow for days below $Q_{80}$ seems to stay at almost the same level in NF for August–October (Fig. 6C).

Further analysis included assessment of low flow magnitude changes, which for the summer and autumn seasons was modeled in HEC-RAS using daily flow hydrographs derived from SWAT. Reach-averaged discharge and flow velocity is presented separately for the main channel and the anabranch. Simulations derived from SWAT indicate that both the discharge rate and mean flow velocity slightly increase in NF (3% for ensemble median) and FF (5% for ensemble median) (Fig. 7). Based on the T-student test, NF changes in discharge and flow velocity were not statistically significant ($p$ value of 0.16 and 0.15 respectively) and the FF changes were statistically significant ($p$ value of 0.0001 and 0.001 respectively).

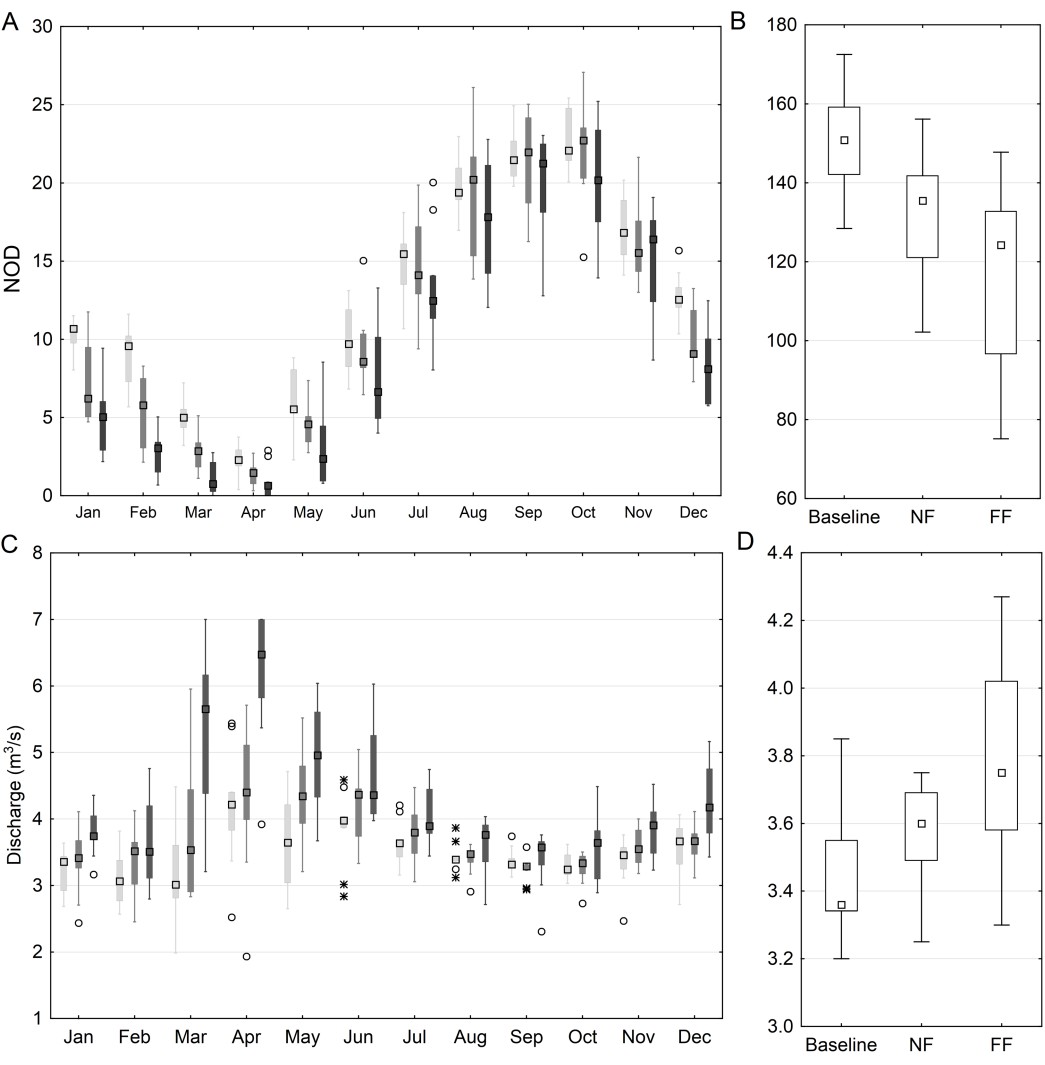

**Figure 6 Mean monthly (A) and annual (B) number of days with low discharge ($Q < Q_{80}$) for the anastomosing reach of the River Narew based on SWAT simulation and the mean monthly (C) and annual (D) value of flow of consecutive days below $Q_{80}$.** NOD stands for number of days, light gray color denotes baseline, medium gray—NF and dark gray—FF horizon; squares denote median values, boxes—1st to 3rd quartile values, whiskers—non outlier range, circles—outliers, asterisks—extremes.

## DISCUSSION

### Anastomosing river perspectives under climate change

The conservation of anastomosing rivers worldwide, although extremely important nowadays, has not been addressed in many studies. The success of their conservation depends on finding and steering the appropriate key factors, which differ from stream to stream and site to site (*Verdonschot & Nijboer, 2002*). *Marcinkowski, Grabowski & Okruszko (2017)* pointed out the low flow to be the main stressor triggering the loss of anastomosing channels in the NNP. In this study, flow conditions were analyzed in terms of duration and magnitude by using a cascade of modeling driven by climate change projections. It was due to the fact that Regional Climate Model (RCM) data operates at

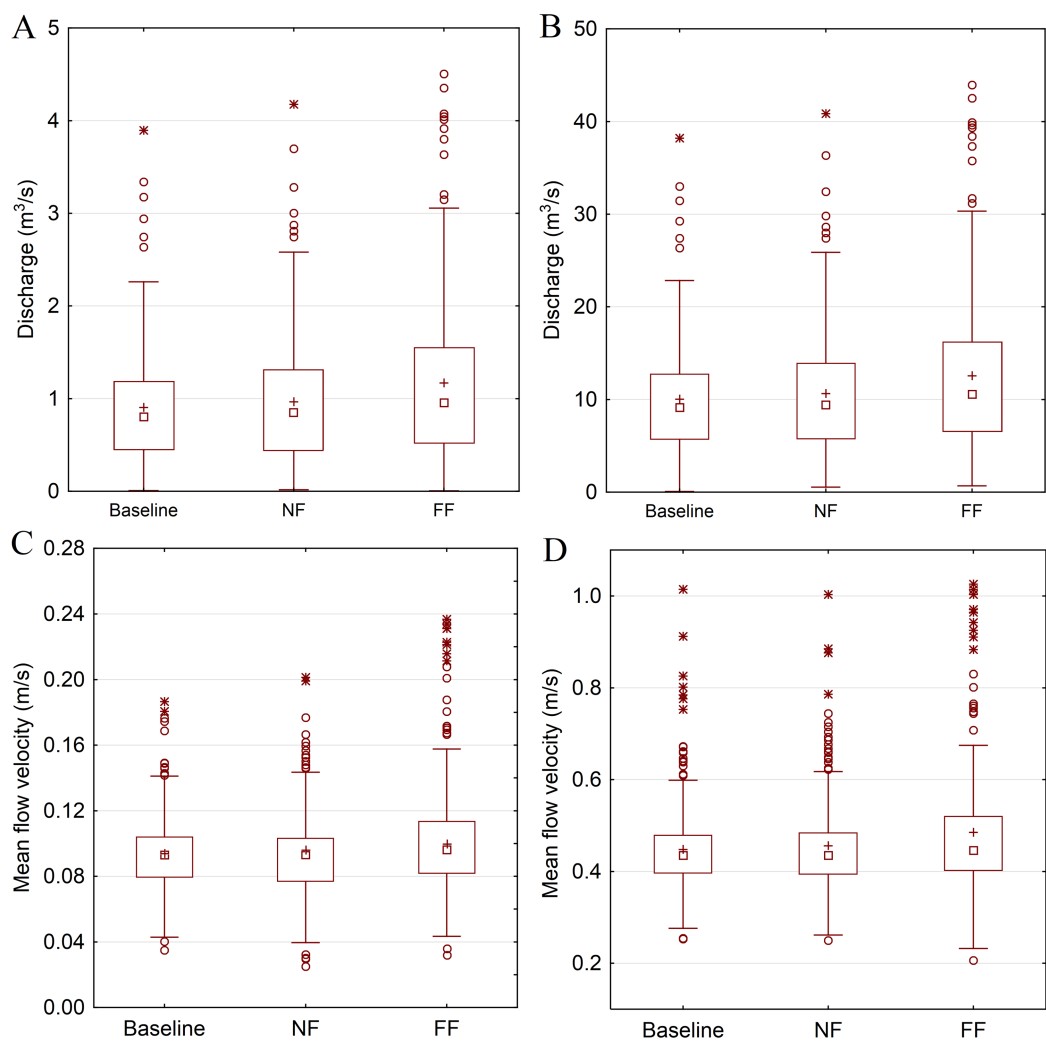

**Figure 7 HEC-RAS model simulations of hydraulic condition changes in the anabranch (A and C) and the main channel (B and D).** Squares denote median values, plus marks—mean values, boxes—1st to 3rd quartile values, whiskers—non outlier range, circles—outliers, asterisks—extremes.

coarse scale, whereas in river conservation programs, particular river reaches of fine scale are targeted. The results were presented at catchment (using SWAT) and reach (using HEC-RAS) scale on an annual and monthly basis. Catchment-averaged mean annual duration of low flows ($Q$ below $Q_{80}$) was projected to decrease and mean flow to increase in both future time horizons. Conclusions drawn on an annual basis results suggest that hydrological conditions might improve in future. However, more detailed monthly analysis indicated that for August–October, when according to *Marcinkowski, Grabowski & Okruszko (2017)* the loss of anastomoses is the most intense, duration of flows below $Q_{80}$ increases and the mean flow in NF is projected to remain at the same level as it is currently. The projected increase in precipitation and flow is distributed mainly to the winter and spring seasons, when vegetation is scarce and the potential of overgrowing the

main cause of channels extinction that is, excessive sediment deposition and plants colonization.

## Study limitations

In this study, we used coupled hydrological and hydraulic models to test the impact of climate change on flow conditions in the anastomosing section of the NNP. The designed methodological approach as well as tools and the datasets used are burdened with some limitations. First of all, it has to be mentioned that low flows, which are of special concern in this study, are mainly triggered by groundwater, whereas both the SWAT and HEC-RAS models do not address this flow component in sufficient detail. To reduce the errors while transferring the projected discharge values from SWAT to HEC-RAS beside the daily stream hydrograph, a groundwater flow (generated by groundwater recharge) from the direct sub-catchment area of the anastomosing section, was added to the HEC-RAS domain. According to *Bates et al. (2008)*, hydrological changes to European rivers caused by climate change are strongly related to the stream flow type. Following their findings for catchments dominated by groundwater flow (Narew case), a decline in groundwater recharge is expected which poses a greater risk of groundwater and surface water drought in these catchments. Therefore, the simplistic approach for groundwater simulation presented in SWAT and HEC-RAS, which might give unreliable results in terms of the magnitude of groundwater flow contribution to the channel, is rather consistent with the overall direction presented by *Bates et al. (2008)*.

## CONCLUSION

The coupled hydrological and hydraulic modeling conducted in this study indicated that low flow conditions in the anastomosing section of the Narew National Park will remain relatively stable in the near future and will slightly improve in far future. Duration of low flows, although projected to decrease on an annual basis, will increase for August–October, when the loss on anastomoses was found to be the most intense. Extremely low flow velocities in the anastomosing arm (<0.1 m/s) nowadays and under future projections seem to remain stable. Such conditions are preferable for in-stream vegetation development as the abundance of macrophytes are typically stimulated at low velocities and their growth is restricted at higher velocities. The perspective of further degradation on the anastomosing system in coming decades requires actions to be undertaken by freshwater ecosystem managers. Our study, although burdened with limitations and uncertainties, might serve as a reference for expected future flow conditions.

### Funding

The authors received no funding for this work.

### Competing Interests

The authors declare that they have no competing interests.

## Author Contributions

- Paweł Marcinkowski conceived and designed the experiments, performed the experiments, analyzed the data, prepared figures and/or tables, authored or reviewed drafts of the paper, and approved the final draft.
- Dorota Mirosław-Świątek conceived and designed the experiments, performed the experiments, authored or reviewed drafts of the paper, and approved the final draft.

## Data Availability

Data is available in the Supplemental Files.

## Supplemental Information

Supplemental information for this article can be found online at http://dx.doi.org/10.7717/peerj.9275#supplemental-information.

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
