# Peer review of "Modelling of climate change impact on flow conditions in the lowland anastomosing river"

_PeerJ, doi:10.7717/peerj.9275_

## Round 0.1 · original submission · Major Revisions

Dear Drs. Marcinkowski and Mirosław-Świątek,

I just received all the reviews of your manuscript. Although all reviewers consider the study very interesting and providing new findings on the topic, some issues need to be considered before the acceptance. Mainly, reviewer#1 find several weaknesses in the model description, reviewer#2 requires more detailed results of in-stream hydrological processes and reviewer#3 is very concerned about how the authors interpret the elevated uncertainty.

Please, consider all comments and suggestions provide by the three reviewers during the revision of your manuscript. All reviewers have included annotated manuscripts in a PDF file. A comprehensive revision of the English of the manuscript is necessary before submitting the new version. Don't forget to include a letter response along with the revised version of the manuscript. In this letter you must respond point by point to each question.

Best regards,
Salva

Reviewer 1 ·

Basic reporting

In general the manuscript reads a correct English.
References are sometimes insufficient but the context is well established (see general comments).
The manuscript follows the structure proposed by the journal.

Experimental design

The manuscript provides original and primary research within the scope of the jounal.
Research questions are well defined but contribution of the article needs to be clarified (see the general comments).
The methods used for the investigation adequately fit the purpose but in the information provided is far from being replicated. Model descriptors are vaguely shown or sometimes not even mentioned (see pdf file).

Validity of the findings

Results are significant to propose recommendations but not irrefutable. The text should read coherent with this.

Additional comments

This paper presents very interesting findings referred to conservation of anastomosing rivers and the use of cascade modelling. It proposes a methodology that combines hydrological and hydraulic models in an original way which may help to predict river conservation problems before these type of rivers disappear from the European continent. The strength of the paper, i.e. the methodology and application to river management, has the potential to be published and fits well with the scope of the journal. However, in order to be published, it needs major modification of its current form, mainly clarifying the methods to be reproducible, focus the discussion on model applications, and rewrite the conclusions to fit with the aims of the paper. In addition, a number of minor comments have been added directly to the attached pdf.
The questions that in my opinion will improve the paper to a publishable level are the following:
1-The introduction creates a well-founded background of the anastomosing river problems. To avoid a “study case paper” it needs to be clarify that the River Narew is just the example where to carry out the investigation. It also needs to state clearer why the paper is taking climate change effects and not human impacts as well (introduced in L37-39).
2-Methods section starts with Site description. It lacks of a good hydrological characterization of the river. River flows, high and low discharges, flooding, seasonality, and other useful data for the reader should be added. Difinition of Low Flow needs clarification (L120-123) and a good referencing. In addition, some more explanation on simulation parameters, boundary conditions, and justification of selected values must be included. In general, this section is poorly explained.
3-Results are simple and clear and well written. There is no mention to the raw supplementary material or explanation of it in the manuscript.
4-Discussion is well divided in two subsections showing the main results of the paper: advantages of the cascade modelling and conservation perspectives of anastomosing rivers. To enhance this section the author must order carefully and clarify the ideas that support the stated hypothesis (see the attached pdf).
5-Conclusions needs to be rewritten completely and does not fit completely tosure that the objectives should fit with conclusions.
It is presented as an abstract and not to highlight the main conclusions of the paper. It is very “case-study” as well. I recommend to state clearly 1) why we should use hydrological models to manage river systems (they may tell different trends than the pure climatic simulations) and 2) why is important to act quickly today in anastomosing river systems (ecological loss in future predictions).
6-Figures and tables need some minor changes (see pdf). Raw data is not mentioned in the main text and there is no explanation on what is actually there. Please provide a key for them.

Annotated reviews are not available for download in order to protect the identity of reviewers who chose to remain anonymous.

Reviewer 2 ·

Basic reporting

no comment

Experimental design

no comment

Validity of the findings

no comment

Additional comments

Modelling of climate change impact on flow conditions in the lowland anastomosing river.

The above manuscript is used an integrated modeling framework of SWAT and HEC-RAS to investigate the hydrodynamics at river reach scale of an anastomosing river system. This study is useful for watershed managers and practitioners to formulate appropriate policy for designing restoration work in the freshwater ecosystem. However, I have following concerns and I want the authors should address them before get published:

1- Please, explain how predicted daily streamflow is lesser in the reach scale while increasing trend in the catchment scale mentioned in the Abstract?
2- Please, include following reference in line 23-24 as additional reference to impacts of climate change to watershed scale:

Giri, S., Arbab, N.N., Lathrop, R.G., 2019. Assessing potential impacts of climate and land use change on water fluxes and sediment transport in a loosely coupled system. Journal of Hydrology 577, 123955.

3- Change to demonstrates line 35.
4- A schematic presentation of Integrated modeling framework representing SWAT, HEC-RAS, climate change inputs, field works as well as output from each model should be provided in a new figure. This figure should be inserted below section 2.3. It will overall provide the modeling framework of the study at a glance to readers. Author can refer different models and input parameters while describing the remaining material and methods section.
5- The low flow threshold value is 7 (m3/sec) based on 80% exceedance probability of daily observed flow. I am curious whether this 80% is randomly assigned or is there a reason behind it?
6- Remove “and” in line 132.
7- Generally, for SWAT calibration Nash Sutcliffe Efficiency is used to calibrate the model. Why did the author use Kling–Gupta efficiency for model calibration? Is there a reason behind it?
8- Line 165 to 173: Authors have described the characteristics of climate change component pattern which should be moved to Results section under “Catchment scale hydrological response to climate change”.
9- Please, provide in the section 2.4.2 climate change scenario regarding how the authors dealt with uncertainty in climate data from the GCM. In the results section, the authors have mentioned that they used “ensemble median approach”. Elaborate that in the section 2.4.2
10- Figure 2 is not refered inside the text in the manuscript. The authors can incorporate in section 2.4.2.
11- Figure 4 does not provide the information written by authors in section 3.1. Authors have mentioned annual and seasonal increase in discharge in the text while the figure is represented in monthly format. I would provide the discharge value on annual and seasonal basis in the graph/ table based on the paragraph written in section 3.1, so that readers can check the figure based on the writing. At current state, its kind of mismatch between writing and figure 4.

11-Similar discripacy was observed for figure 3 and writing of annual temperature and precipitation as well as their seasonal pattern in section 2.4.2

12- One of the major piece of information missing in this manuscript is “ how did the authors model the reaches starting from section NG and splitting into “NR” and “NL” and back to “ND” (figure 1C) in SWAT model?. Does all the reached comes under one subbasin or all the reaches are in different subbasins. It would be more clear If author can provide subbasin background in figure 1C. The other question I have is if the reaches of NR and NL comes under 1 subbasin, then how did authors get the streamflow value for two reaches in 1 subbasin as SWAT model assigns only 1 reach per subbsin.

The uniqueness of this study is understanding the in-stream hydraulics and morphology of anastomosing river systems. Therefore, authors need to describe in more detail how they developed that section in SWAT model. This will help other researchers to model the similar river systems around the world.
13- Provide a paragraph of your study limitation

Annotated reviews are not available for download in order to protect the identity of reviewers who chose to remain anonymous.

Reviewer 3 ·

Basic reporting

This is an overall interesting and well-structured manuscript on the impacts of climate change on an anastomosing river (reach) in the Narew river basin in Poland. The authors present an integrated modeling framework using SWAT as a baseline catchment hydrology model that provides the driving runoff data for the HEC-RAS model, applied to a selection of river reaches, where degradation has been observed in previous studies. The study applies bias-corrected data from nine EURO-CORDEX simulations for the baseline, near future and far future time horizons under the RCP4.5-scenario.

Findings indicate a fairly small impact on low flow conditions, yet with an assumed trend towards slightly decreasing flow velocities and an increased number of lof flow days. It is argued that the sign of change requires a consideration of climate change impacts to increase the preparedness level for timely and appropriate conservation measures for river (reach) protection.

The manuscript is generally will written and clearly structured; the setup is logical, the conducted work done and presented in a straight-forward manner..

Still, I have some concerns that I would consider to require "major revisions" before the manuscript merits publication in PeerJ. I made many comments in the annotated pdf, which I kindly ask you to thoroughly take into consideration.

Experimental design

no comment

Validity of the findings

These are my major points in summary:

Language of the manuscript is generally good, but there are substantial problems in the usage of definite and indefinite articles. I made several corrections but then gave up - there are so many... Please revise the whole manuscript for this particular issue as it makes some sentences very difficult to comprehend. Further, I recommend to cut down on the number of self-citations, especially in the introduction (lines 48ff) or the discussion chapter 4.2 (lines 304ff).

Please help to cure my concerns in these contextual aspects:

- comment on the general problem of investigating low flow with a modeling approach that doesn't explicitly consider groundwater flow in a substantial matter. The fact that SWAT has been applied in many several contexts is not a very satisfactory explanation.

- comment on the mismatch of results from SWAT and HEC-RAS. Why are the signs of flow change in the models different? Which ones are correct??

- can you really speak of an IMF, as long as the applied models are uni-directionally coupled?

- most importantly: can you really claim significant findings of change, given the very limited (and partly even contradictory) range of changes captured in the modelling framework? You should comment more on the inherent uncertainties in your setup and then balance your statements against these uncertainties.

- given the rather small impacts on flow velocity and volumes, I wonder whether you are not overemphasizing the statements towards the impacts on the anastomosing river and the consequential conservation needs?

Additional comments

no comment

Annotated reviews are not available for download in order to protect the identity of reviewers who chose to remain anonymous.

---

## Round 0.2 · accepted · Accept

Dear Dr. Marcinkowski,

I just want to inform you that the changes made to your manuscript have substantially improved your work and it is now acceptable to be published in PeerJ.

Congratulations!

Salva

Reviewer 2 ·

Basic reporting

The manuscript is now clear and authors have used good number of literature to support this research. The structures figures, and tables are articulated clearly.

Experimental design

The research question is well defined and relevant to the real world scenarios. The study methods are described sufficiently and are based on science based information.

Validity of the findings

Conclusions are modified based on the suggestions from the reviewers.

Additional comments

Authors have addressed my comments as well as comments from other reviewers. I am satisfied with the effort and recommend for publication.